# ADVERSARIAL AUDIO SUPER-RESOLUTION WITH UNSUPERVISED FEATURE LOSSES

## ABSTRACT

Neural network-based methods have recently demonstrated state-of-the-art results on image synthesis and super-resolution tasks, in particular by using variants of generative adversarial networks (GANs) with supervised feature losses. Nevertheless, previous feature loss formulations rely on the availability of large auxiliary classifier networks, and labeled datasets that enable such classifiers to be trained. Furthermore, there has been comparatively little work to explore the applicability of GAN-based methods to domains other than images and video. In this work we explore a GAN-based method for audio processing, and develop a convolutional neural network architecture to perform audio super-resolution. In addition to several new architectural building blocks for audio processing, a key component of our approach is the use of an autoencoder-based loss that enables training in the GAN framework, with feature losses derived from unlabeled data. We explore the impact of our architectural choices, and demonstrate significant improvements over previous works in terms of both objective and perceptual quality.

## 1  INTRODUCTION

Deep convolutional neural networks (CNNs) have become a cornerstone in modern solutions for image and audio analysis. Such networks have excelled at supervised discrimination tasks, for instance on ImageNet (Deng et al., 2009; Simonyan & Zisserman, 2014), where image classifier networks are trained on a large corpus of labeled data. More recently, CNNs have successfully been applied to data synthesis problems in the context of generative adversarial networks (GANs) (Goodfellow et al., 2014). In the GAN framework, a neural network is used to synthesize new instances from a modeled distribution, or resolve missing details given lossy observations. In the latter case, the GANs have been shown to greatly improve reconstruction of fine texture details for images, compared to standalone sample-space losses that result in overly smoothed outputs (Dosovitskiy & Brox, 2016; Isola et al., 2017; Ledig et al., 2017). However, GANs are notoriously hard to train, and the use of conventional sample-space objectives in conjunction with an adversarial loss either de-stabilizes training, or results in outputs with significant artifacts (Figure 1).

To address the smoothness problem described above, previous works typically augment or replace conventional sample-space losses with a *feature loss* (also called a perceptual loss) (Dosovitskiy & Brox, 2016; Ledig et al., 2017; Johnson et al., 2016). Instead of distance in raw sample-space, such feature losses reflect distance in terms of the feature maps of an auxiliary neural network. While classifier-based feature losses are effective, they require either a pre-trained neural network that is applicable to the problem domain (e.g., synthesizing images of cats), or a labeled dataset that is amenable to training a relevant classifier. Training new classifiers for use in a feature loss can be non-trivial for numerous reasons. Besides the difficulty of training large classifiers that are commonly used for feature losses, such as VGG (Simonyan & Zisserman, 2014), creating a labeled dataset that is sufficiently large and diverse is often infeasible.

In this work, we sidestep the difficulty of training auxiliary classifiers by developing a feature loss that is unsupervised. In particular, we focus on an audio modeling task called super-resolution, where the goal is to generate high-quality audio given down-sampled, low-resolution input. Inspired by previous work on audio and image super-resolution, we develop a neural network architecture for

---

Audio samples available: https://mugandemo.github.io/mugandemo/

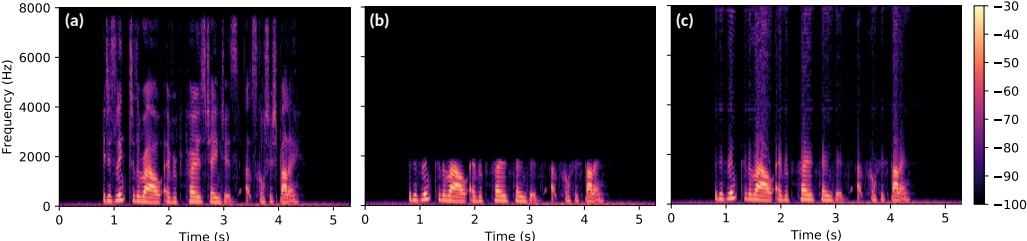

Figure 1: **(a)** High-resolution, **(b)** low-resolution, **(c)** and super-resolution spectrograms. The super-resolution spectrogram corresponds to audio generated by the GAN described in Section 3, trained only with an adversarial loss and conventional $L2$ loss. Training tends to either diverge, or results in audio with persistent high-frequency tones and alias-like artifacts.

end-to-end super-resolution that operates on raw audio. In addition to providing new algorithms to model audio, our work suggests new techniques to improve GAN-based methods in other domains such as images and video. Specifically, our contributions are as follows:

1. We formulate a new, general-purpose feature loss that is fully unsupervised and circumvents the need for problematic classifier-based models.

2. We successfully adapt the adversarial framework for audio processing, and provide solutions to previously unsolved problems associated with the application of GANs to audio.

3. We demonstrate our methods in an end-to-end architecture for audio super-resolution, with state-of-the-art results on both speech and music tasks.

4. We analyze important architectural parameters of our model, and in particular discover previously-unobserved behavior with effective receptive field sizes.

## 2 BACKGROUND & RELATED WORK

**Audio super-resolution**  Audio super-resolution is the task of constructing a high-resolution audio signal from a low-resolution signal that contains a fraction of the original samples. Concretely, given a low-resolution sequence of audio samples $x_l = (x_{1/R_l}, \ldots, x_{R_l T/R_l})$, we wish to synthesize a high-resolution audio signal $x_h = (x_{1/R_h}, \ldots, x_{R_h T/R_h})$, where $R_l$ and $R_h$ are the sampling rates of the low and high-resolution signals, respectively. We denote $R = R_h/R_l$ as the *upsampling ratio*, which ranges from 2 to 6 in this work. Thus, the audio super-resolution problem is equivalent to reconstructing the missing frequency content between frequencies $R_l/2$ and $R_h/2$.

There is a vast body of prior work on audio super-resolution in the signal and audio processing communities under the term *artificial bandwidth extension* (Larsen & Aarts, 2004). Neural network-based methods in this domain generally apply a DNN on top of hand-crafted features as part of complex bandwidth extension systems (Liu et al., 2015; Abel & Fingscheidt, 2018). Gaussian mixture and hidden Markov models have also been used (Bachhav et al., 2017; Tokuda et al., 2013), but these methods generally perform worse compared to neural networks (Abel & Fingscheidt, 2018). In contrast with the works above, our method does not rely on hand-crafted features (e.g., transformations or cepstrum coefficients), and is not specific to problems in speech modeling.

**Audio modeling with neural networks**  Learning-based approaches for audio have also been explored in the largely in the context of representation learning, generative modeling, and text-to-speech (TTS) systems. Unsupervised methods such as convolutional deep belief networks (Lee et al., 2009) and bottleneck CNNs (Aytar et al., 2016) have been shown to learn useful representations from audio, such as phonemes and sound textures. Stacked autoencoders (Vincent et al., 2010) and variational autoencoders (Kingma & Welling, 2014; Sonderby et al., 2016) have been used for denoising, image generation, and music synthesis (Sarroff & Casey, 2014). Bottleneck-like CNNs have also demonstrated significant improvements for audio super-resolution in supervised settings compared to previous DNN and spline-based methods (Kuleshov et al., 2017). Donahue et al. (2018) is among the first to develop methods for raw audio synthesis with GANs. Notably, Donahue et al.

(2018) show that non-trivial modifications of GAN architectures are required to generative diverse and plausible audio outputs. We build on the works above by developing a GAN framework for audio super-resolution with an improved bottleneck-style generator, and show that leveraging representations learned from unsupervised training greatly aid the super-resolution task. Autoregressive probabilistic models have recently demonstrated state-of-the-art results for generation of music (Engel et al., 2017), general audio (van den Oord et al., 2016; Mehri et al., 2017), and for parametric TTS systems (Sotelo et al., 2017). Several works have leveraged model distillation (van den Oord et al., 2018; ClariNet, 2019)[1] to mitigate the overhead of autoregressive methods, making them feasible for real-time audio generation. In general, our work can be used to augment existing speech synthesis systems, including those that employ autoregressive methods. For instance, the unsupervised feature loss proposed in our work could be used as a drop-in replacement for the classifier-based feature loss used by van den Oord et al. (2018). While we are not aware of any efforts that explore autoregressive modeling for audio super-resolution, we believe it may be a promising future direction.

**Generative adversarial networks for images**  Generative methods have been extensively explored for image generation and super-resolution. Building upon the original formulation of Goodfellow et al. (2014), GANs have been continuously improved to generate plausible, high-fidelity images (Radford et al., 2015; Denton et al., 2015; Berthelot et al., 2017; Karras et al., 2018). GAN variants conditioned on class labels or object sketches have also demonstrated promising results on tasks such as in-painting and style transfer (Mirza & Osindero, 2014; Isola et al., 2017).

## 3 METHOD

**GANs for Super-Resolution**  GANs developed for super-resolution tasks have several important differences compared to the original formulation of Goodfellow et al. (2014). When used to generate new instances from a data distribution $p_{\text{data}}$, the generator $(G)$ parameterized by $\theta_G$ learns the mapping to data space as $G(z; \theta_G)$, where $z$ is a latent noise prior. The discriminator $(D)$ parameterized by $\theta_D$ then estimates the probability that $G(z; \theta_G)$ was drawn from $p_{\text{data}}$ rather than the generator distribution $p_g$. In contrast, for super-resolution, $G$ is no longer conditioned on noise and learns the mapping to high-resolution data space $p_h$ as $G(x_l; \theta_G)$, where $x_l$ is drawn from the low-resolution data distribution $p_l$. The task of $D$ is to discriminate between samples from the high-resolution and super-resolution (generator) distributions $p_h$ and $p_g$, respectively. Since low-resolution data $x_l$ corresponds directly to a downsampled version of $x_h$ during training, we expect $G(x_l; \theta_G) \approx x_h$. $G$ and $D$ are optimized according to the two-player minimax problem:

$$\min_{\theta_G} \max_{\theta_D} \mathbb{E}_{x_h \sim p_h(x_h)} \left[ \log D\left(x_h; \theta_D\right) \right] + \mathbb{E}_{x_l \sim p_l(x_l)} \left[ \log(1 - D\left(G\left(x_l; \theta_G\right)\right)) \right] \tag{1}$$

This framework enables the joint optimization of two neural networks - $G$ generates super-resolution data with the goal of fooling $D$, and $D$ is trained to distinguish between real and super-resolved data. Thus, the GAN approach encourages $G$ to learn solutions that are hard to distinguish from real, high-resolution datum.

**Architecture overview**  MU-GAN (Multiscale U-net GAN) is composed of three models that all operate on raw audio[2] - a generator $(G)$, discriminator $(D)$, and convolutional autoencoder $(A)$ (Figure 2). The generator's task is to learn the mapping between the low and high-resolution data spaces, corresponding to signals $x_l$ and $x_h$, respectively. The discriminator's task is then to classify whether presented data instances are real, or produced by the generator. In addition to $G$ and $D$, the autoencoder extracts perceptually-relevant features from both real and super-resolved data for use in feature-space loss functions. The use of $A$ is crucial in the GAN framework, as generators trained solely on L2 or other sample-space losses suffer from training instability or output artifacts (Ledig et al., 2017).

**Multiscale convolutional layers**  In comparison to images, audio signals are inherently periodic with time-scales on the order of 10's to 100's of samples. As a consequence, filters with very large

---

[1] https://openreview.net/forum?id=HklY120cYm
[2] Data is encoded in 32-bit floating point format.

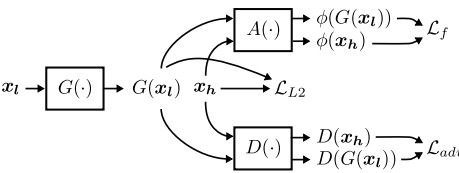

Figure 2: Overview of the model architecture and corresponding loss terms.

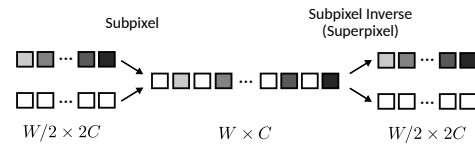

Figure 3: Subpixel and superpixel layers for increasing and decreasing spatial resolution, respectively.

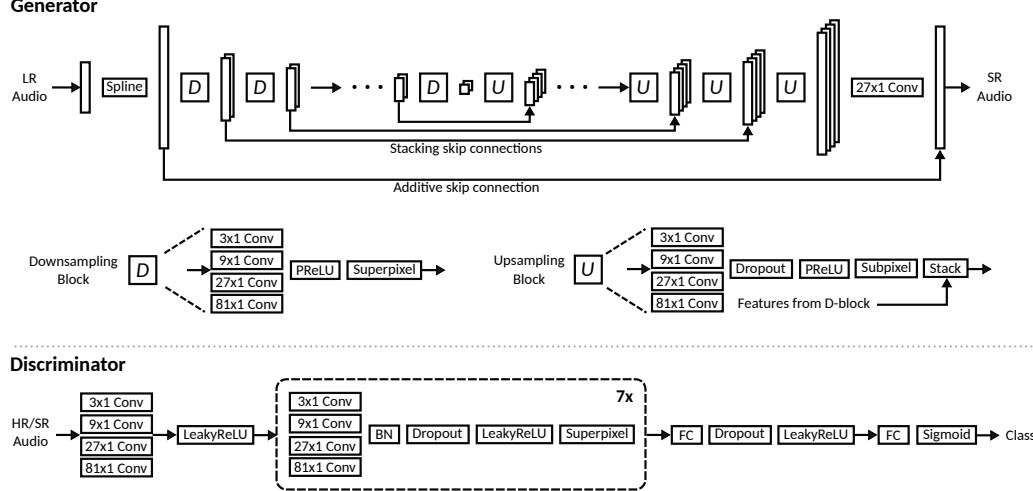

Figure 4: Generator and discriminator models.

receptive fields are required to create high quality raw audio (Donahue et al., 2018; van den Oord et al., 2016). Previous work with classifier models also suggests that varying the filter size within a network helps capture information at multiple scales (Szegedy et al., 2015). Leveraging these observations, we use a multiscale convolutional building block composed of concatenated 3x1, 9x1, 27x1, and 81x1 filters. In practice, and with a fixed number of parameters for a given layer, we found that filters larger than 81x1 provided no additional benefit, while omitting large filter sizes resulted in significantly degraded audio quality. We interpret the poor performance of small filters as being a byproduct of their frequency selectivity; it is well known from signal processing theory that the resolution of an FIR filter's frequency response is proportional to the length of the filter.

**Superpixel layers**   Recently, it has been shown that pooling and strided convolutions tend to induce periodic "checkerboard" artifacts (Odena et al., 2016; Donahue et al., 2018). Shi et al. (2016) developed a *subpixel* layer to increase spatial resolution, and showed that it is less prone to checkerboard artifacts. While the subpixel layer was subsequently adopted by several works (Kuleshov et al., 2017), no efforts have evaluated the performance of the inverse operation for *decreasing* spatial resolution. Concretely, the inverse subpixel operator interleaves samples from the time dimension into the channel dimension, and thus reduces the spatial resolution by an integer factor. We refer to this simple inverse operation as a *superpixel* layer (Figure 3), and use it as a drop-in replacement for strided convolution and pooling layers.

**Generator network**   The high-level architecture for the generator network (Figure 4, top) is inspired by autoencoder-like U-net models (Ronneberger et al., 2015; Isola et al., 2017; Kuleshov et al., 2017). In a U-net-style model, the first half of the network consists of $B$ downsampling blocks (D-blocks) that perform feature extraction at multiple scales and resolutions [3]. The second half the model consists of $B$ upsampling blocks (U-blocks), which successively increase the spatial resolution of the signal. We use multi-scale convolutional layers throughput the generator network, and replace all strided convolutions with superpixel layers.

---

[3]Note that to have matching resolutions at the input and output of $G$, the LR signal is first upsampled with a cubic spline.

**Discriminator network**    The discriminator (Figure 4, bottom) is used during training to differentiate between real, high-resolution audio and super-resolved signals produced by the generator. Our design is loosely based on the recommendations of Radford et al. (2015), and the image discriminator from Ledig et al. (2017). All discriminator activations are LeakyReLU (Maas et al., 2013) with $\alpha = 0.2$. As with the generator, we use multi-scale convolutions, and the superpixel layer described above instead of strided convolutions to minimize artifacts in the loss gradients (Odena et al., 2016).

**Autoencoder network**    The autoencoder $A$ is used to extract perceptually relevant features from the low and high-resolution signals. The features extracted by $A$ are incorporated in the generator's feature loss $\mathcal{L}_f$, which is described in more detail in following sections. For the specific implementation of $A$, we use a modified version of the generator model that excludes all additive and stacking skip connections. Hence, the model for $A$ is a convolutional autoencoder, augmented with multiscale convolutional layers, and super/subpixel layers for down/up-sampling.

**Loss functions**    MU-GAN incorporates several loss terms for training the generator and discriminator. The first term in the generator loss is the sample-space L2 loss, given by[4]

$$\mathcal{L}_{L2} = \frac{1}{W} \sum_{i=1}^{W} \|\boldsymbol{x}_{\boldsymbol{h},i} - G(\boldsymbol{x_l})_i\|_2^2. \tag{2}$$

We found that using only the sample-space and adversarial losses either resulted in little to no improvement over the baseline non-GAN model, or introduced persistent audible artifacts (e.g., high-frequency tones, Figure 1). These findings are in line with those of Ledig et al. (2017), who experience similar issues with images [5]. As described in Section 2, the use of a feature loss with GAN training encourages the generator to learn solutions that incorporate perceptually relevant texture details. Given the autoencoder $A$, we denote the output feature tensor at the bottleneck of the autoencoder as $\phi$. The feature loss $\mathcal{L}_f$ is then given by

$$\mathcal{L}_f = \frac{1}{C_f W_f} \sum_{c=1}^{C_f} \sum_{i=1}^{W_f} \|\phi(\boldsymbol{x_h})_{i,c} - \phi(G(\boldsymbol{x_l}))_{i,c}\|_2^2, \tag{3}$$

where $W_f$ and $C_f$ denote the width and channel dimensions for the feature maps of autoencoder bottleneck. The adversarial loss $\mathcal{L}_{adv}$ is determined by discriminator's ability to discern whether data produced by the generator is real or fake. We use the gradient-friendly formulation originally posed in Goodfellow et al. (2014), given by

$$\mathcal{L}_{adv} = -\log D(G(\boldsymbol{x_l})). \tag{4}$$

The composite loss $\mathcal{L}_G$ for the generator is then given by the sum of the losses above, and the discriminator loss $\mathcal{L}_D$ derives directly from the GAN optimization objective in Equation 1, i.e.,

$$\mathcal{L}_G = \mathcal{L}_{L2} + \lambda_f \mathcal{L}_f + \lambda_{adv} \mathcal{L}_{adv}, \tag{5}$$

$$\mathcal{L}_D = -\left[\log D(\boldsymbol{x_h}) + \log(1 - D(G(\boldsymbol{x_l}))) \right], \tag{6}$$

where $\lambda_f$ and $\lambda_{adv}$ are constant scaling factors.

## 4    EXPERIMENTS

**Datasets**    We evaluate our methods on three super-resolution tasks derived from the VCTK Corpus (Yamagishi), and the non-vocal music dataset from Mehri et al. (2017). For speech from VCTK, we compose a dataset with recordings from a single speaker (the *Speaker1* task), and a dataset

---

[4] We write losses with respect to a single sample, with an implicit mean over the minibatch dimension.

[5] Ledig et al. (2017) posit that this poor performance is due to the competing nature of the adversarial and sample-space losses.

with recordings from multiple speakers (the *Speaker99* task). *Speaker1* consists of the first 223 recordings from VCTK speaker 225 for training, and the final 8 recordings for testing. *Speaker99* uses all recordings from the first 99 VCTK speakers for training, and recordings from the last 10 speakers for testing. *Piano* uses the standard 88%-6%-6% train/validation/test split. For all tasks, the dataset is created by first applying an anti-aliasing lowpass filter, and then sampling random patches of fixed length from the resulting audio. Note that for the sake of direct comparison, the datasets above are the same as those used in Kuleshov et al. (2017).

**Training methodology**   For *Speaker1*, we instantiate variants of MU-GAN and train for 400 epochs. For the larger datasets *Speaker99* and *Piano*, models are trained for 150 epochs. The epoch number is empirically selected based on observed convergence, and performance saturation on the validation set. For all models, we use the ADAM optimizer with learning rate 1e-4, $\beta_1 = 0.9$, $\beta_2 = 0.999$, and a batch size of 32. For the autoencoder feature losses, we instantiate a model with $L = 4$, and train for 400 epochs on the same dataset as its associated GAN model. The loss scaling factors $\lambda_f$ and $\lambda_{adv}$ are fixed at 1.0 and 0.001, respectively. Additional details on model hyperparameters can be found in Appendix A.1.

**Performance metrics**   We use three metrics to assess the quality of super-resolved audio: (1) signal-to-noise ratio (SNR), (2) log-spectral distance (LSD), and (3) mean opinion score (MOS). The SNR is a standard metric in signal processing communities, defined as

$$\text{SNR}\,(x, x_{ref}) = 10 \log_{10} \frac{\|x_{ref}\|_2^2}{\|x - x_{ref}\|_2^2}, \tag{7}$$

where $x$ is an approximation of reference signal $x_{ref}$. LSD (Gray & Markel, 1976) measures differences between signal frequencies, and has better correlation with perceptual quality compared to SNR (Jie et al., 2014; Kuleshov et al., 2017). Given short-time discrete Fourier transforms $X$ and $X_{ref}$, the LSD is given by

$$\text{LSD}\,(X, X_{ref}) = \frac{1}{W} \sum_{w=1}^{W} \sqrt{\frac{1}{K} \sum_{k=1}^{N} \left( \log_{10} \frac{|X(w,k)|^2}{|X_{ref}(w,k)|^2} \right)^2}, \tag{8}$$

where $w$ and $k$ are the window and frequency bin indices, respectively[6]. Perceptual evaluation of speech quality (PESQ) (ITU-T, 2001) is an industry-standard methodology for the assessment of speech communication systems. Given reference and degraded audio signals, PESQ models the mean opinion score (MOS) of a group of listeners. Specifically, we use PESQ to produce MOS-LQO (listening quality objective) scores (ITU-T, 2003), which range from 1 to 5.

**Impact of superpixel layers**   We find that the use of superpixel layers results in ∼14% improvement in training time across model sizes, with insignificant differences in terms of objective quality metrics. Differences in audio produced by the two methods were also imperceptible in informal self-blinded listening tests. This indicates that superpixel layers may be a suitable replacement for conventional strided convolutions, while offering improvements in training time without performance loss. Additional details are provided in Appendix A.2.

**Objective performance evaluation**   Table 1 shows the quantitative performance of MU-GAN against other recent works. We denote *MU-GAN8* as an instance of MU-GAN with a depth parameter of $L = 8$, i.e., with 8 downsampling and 8 upsampling blocks. *U-net4* is the model with $L = 4$ from Kuleshov et al. (2017). To eliminate depth as a factor in the performance comparison, we reimplement the architecture from Kuleshov et al. (2017) with $L = 8$, denoted as *U-net8*.

Table 1 shows that *MU-GAN8* often performs worse in terms of SNR compared to the baseline models, but has lower LSD and higher MOS-LQO. This indicates that while *MU-GAN8* produces reconstructions with lower SNR, deviating in terms of sample-wise distance results in synthesis of more perceptually-relevant frequency content. The exception is with the *Piano* task, where *MU-GAN8* performs orders of magnitude better than the U-net baseline in terms of SNR. In general, we

---

[6]We use non-overlapping Fourier transform windows of length 2048.

Table 1: Objective comparison with baseline super-resolution networks[†].

| | | Up. Ratio $R = 2$ | | | Up. Ratio $R = 4$ | | | Up. Ratio $R = 6$ | | |
| | | U-net4 | U-net8 | MU-GAN8 | U-net4 | U-net8 | MU-GAN8 | U-net4 | U-net8 | MU-GAN8 |
|---|---|---|---|---|---|---|---|---|---|---|
| *Speaker1* | SNR | 21.1 | **21.94** | 21.40 | 17.1 | **18.68** | 17.72 | 14.4 | **14.85** | 13.98 |
| | LSD | 3.2 | 2.24 | **1.63** | 3.6 | 2.34 | **1.92** | 3.4 | 2.92 | **1.95** |
| | MOS-LQO | - | 4.54 | **4.54** | - | **3.81** | 3.79 | - | 2.97 | **3.21** |
| *Speaker99* | SNR | **20.7** | 20.05 | 20.01 | **16.1** | 14.30 | 14.03 | 10.0 | **11.11** | 10.92 |
| | LSD | 3.1 | 2.22 | **2.14** | 3.5 | 2.92 | **2.72** | 3.7 | 3.23 | **2.97** |
| | MOS-LQO | - | 3.68 | **3.75** | - | 2.68 | **2.93** | - | 2.44 | **2.69** |
| *Piano* | SNR | 30.1 | 44.98 | **52.03** | 23.5 | 31.71 | **32.28** | 16.1 | 22.53 | **24.71** |
| | LSD | 3.4 | 1.12 | **0.90** | 3.6 | 1.35 | **1.30** | 4.4 | 1.53 | **1.41** |

[†] Metrics for *U-net4* are taken directly from Kuleshov et al. (2017); those for *U-net8* are from our reimplementation.

Table 2: A/B test user study scores.

| | Piano | | Speaker99 | | | |
| | #1 | #2 | #1 | #2 | #3 | #4 |
|---|---|---|---|---|---|---|
| *MU-GAN8* | **9** | **15** | **14** | **11** | **15** | **10** |
| *U-net8* (baseline) | 5 | 3 | 4 | 6 | 4 | 8 |
| No preference | 8 | 4 | 4 | 5 | 3 | 4 |

$R = 4$: *Piano* #1, *Speaker99* #1, #3
$R = 6$: *Piano* #2, *Speaker99* #2, #4

Table 3: *Speaker1* objective metrics for *MU-GAN8* trained with the speech classifier-based loss, and proposed loss ($\mathcal{L}_{f,SV}$, $\mathcal{L}_f$).

| | $R = 2$ | | $R = 4$ | | $R = 6$ | |
| | $\mathcal{L}_{f,SV}$ | $\mathcal{L}_f$ | $\mathcal{L}_{f,SV}$ | $\mathcal{L}_f$ | $\mathcal{L}_{f,SV}$ | $\mathcal{L}_f$ |
|---|---|---|---|---|---|---|
| SNR | 21.28 | 21.40 | 17.57 | 17.72 | 13.85 | 13.98 |
| LSD | 1.65 | 1.63 | 1.92 | 1.92 | 1.99 | 1.95 |
| MOS-LQO | 4.54 | 4.54 | 3.67 | 3.79 | 3.24 | 3.21 |

also find that performance on the speech tasks generally saturates at $R = 2$ for both *U-net8* and *MU-GAN8*. Informal listening tests confirm that there are minimal differences at $R = 2$, indicating that more difficult up-sampling ratios (i.e., $R = 4, 6$) are better suited for grounds of further comparison.

**Subjective quality analysis** To evaluate the performance of *MU-GAN* with real listeners, we perform a randomized, single-blinded user study with 22 participants (Table 2). The study presents pairs of audio clips produced by *MU-GAN8* and the best baseline model *U-net8*, and asks participants to select a preferred clip, or "No preference." We present two clips from *Piano*, and four sonically diverse clips from *Speaker99*. Table 2 shows that in all cases, listeners prefer audio produced by *MU-GAN8* over the baseline method.

In general, we observe that audio produced by MU-GAN has greater clarity compared to audio produced by the baseline networks. The quality difference is most apparent during consonant sounds, which have more high-frequency content compared to typical vowel sounds. For instance, in the phrase "Ask her to bring these things from the store," (Figure 5, bottom row) the consonant sounds in 'Ask,' 'things,' and 'store' have noticeably better articulation. In contrast, audio from the best baseline, *U-net8*, sounds relatively dull and "muffled" in comparison. Note that on some audio clips super-resolved at $R = 4, 6$, we observe intermittent, audible noise that is not present in the baseline reconstructions. To ensure a fair analysis, we include audio clips in the user study where this noise is apparent. Additional comments on spurious noise can be found in Appendix A.3.

**Comparison with classifier-based feature loss** We compare the proposed unsupervised feature loss with the classifier-based loss from Germain et al. (2018) [7]. Germain et al. (2018) use a VGG-based (Simonyan & Zisserman, 2014) network as a feature loss for speech denoising, and train the loss network on classification and audio tagging tasks from DCASE 2016 (Mesaros et al., 2018). Table 3 shows objective results obtained using the proposed loss $\mathcal{L}_f$ and the classifier-based loss $\mathcal{L}_{f,SV}$ on the *Speaker1* task. Across all up-sampling ratios, the proposed unsupervised method performs on-par (and slightly better in some cases) compared to the classifier-based loss. Thus, our results suggest that using a domain-specific classifier-based loss may not provide any advantage in terms of performance. Given the issues related to training classifier models on general audio (Section 1), our method may be an attractive solution that does not compromise audio quality.

---

[7] The authors' pre-trained classifier models are obtained from
https://github.com/francoisgermain/SpeechDenoisingWithDeepFeatureLosses

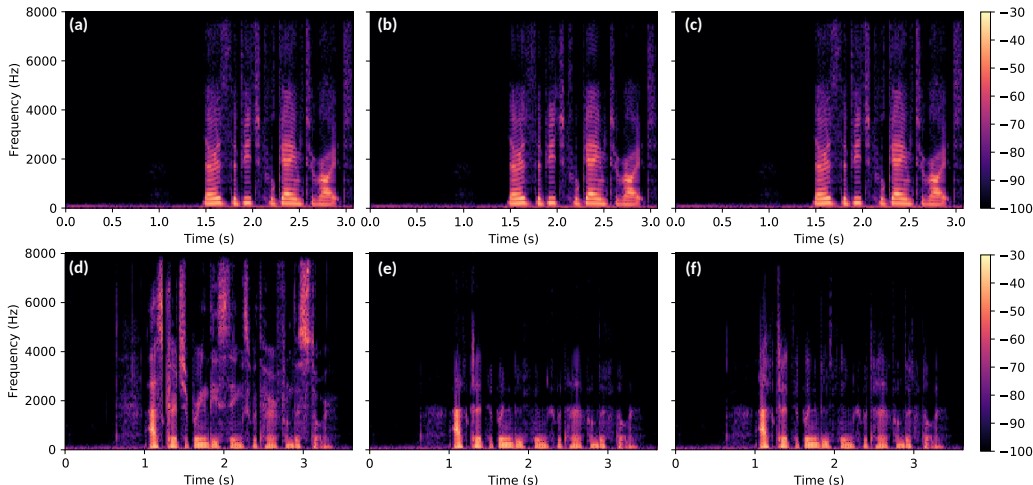

Figure 5: Spectrograms from the *Speaker1* task at $R = 2$ (top row, Speaker 225), and *Speaker99* task at $R = 4$ (bottom row, Speaker 360). **(a-d)** high-resolution, **(b-e)** super-resolved with *U-net8*, and **(c-f)** super-resolved with *MU-GAN8*. Increased synthesis of high-frequency content by MU-GAN8 becomes more pronounced at difficult up-sampling ratios.

Table 4: MOS-LQO for ablated models on the *Speaker1* task.

| | Configuration | | | | | |
|---|---|---|---|---|---|---|
| Up. Ratio | *MU-GAN4* $-\mathcal{L}_f - \mathcal{L}_{adv}$ | *MU-GAN4* $-\mathcal{L}_{adv}(+\mathcal{L}_f)$ | *MU-GAN4* $(+\mathcal{L}_{adv} + \mathcal{L}_f)$ | *MU-GAN8* $-\mathcal{L}_f - \mathcal{L}_{adv}$ | *MU-GAN8* $-\mathcal{L}_{adv}(+\mathcal{L}_f)$ | *MU-GAN8* $(+\mathcal{L}_{adv} + \mathcal{L}_f)$ |
| $R = 2$ | 3.53 | 4.54 | 4.54 | 4.54 | 4.54 | **4.54** |
| $R = 4$ | 3.15 | 3.55 | 3.62 | 3.74 | 3.79 | **3.79** |
| $R = 6$ | 2.78 | 3.07 | 3.12 | 3.15 | 3.17 | **3.21** |

**Ablation Analysis** Table 4 shows the MOS-LQO metrics for the MU-GAN architecture with ablated model parameters. While almost all variations perform similarly well at $R = 2$, adding depth (i.e., from $L = 4$ to $L = 8$) and additional loss terms improves performance on harder up-sampling ratios. Furthermore, adding the adversarial loss and unsupervised feature loss terms improve MOS-LQO monotonically. For *MU-GAN8*, we see diminishing returns from adding the additional loss terms; much of the improvement over *MU-GAN4* appears to come from the additional depth. On the other hand, adding the $\mathcal{L}_f$ and $\mathcal{L}_{adv}$ losses to the *MU-GAN4* variant yields significant benefits, such that its performance is comparable to that of *MU-GAN8*. This indicates that the feature and adversarial losses may be particularly useful to mitigate underfitting, or to decrease model size iso-performance.

**Receptive Field Analysis** While previous works have stressed the importance of large receptive fields (van den Oord et al., 2016; Luo et al., 2016; Yu & Koltun, 2016), little work exists to quantify receptive field sizes on practical problems. We use the methods of Luo et al. (2016) to measure the *effective receptive field* (ERF)[8] of our model on different tasks with varying architectural hyperparameters. Notably, we find that while the theoretical receptive field of our network is on the order of hundreds of milliseconds (thousands of samples), measured ERF's are generally no wider than 100 samples. Furthermore, we find that ERF size is strongly correlated with problem difficulty, rather than architectural hyperparameters such as depth and specific loss terms. While Luo et al. (2016) found that ERF size always increased compared to the ERF at initialization, we find that ERF *decreases* in many cases (Figure 6). Our findings imply that there may be subtle but important tradeoffs involved with receptive field size (for instance, ERF size versus noise rejection), and suggest promising avenues for deeper investigation of architectures that rely on large receptive fields.

---

[8]As in Luo et al. (2016), we define samples that lie within $2\sigma$ of the gradient-magnitude sample mean as "within-ERF."

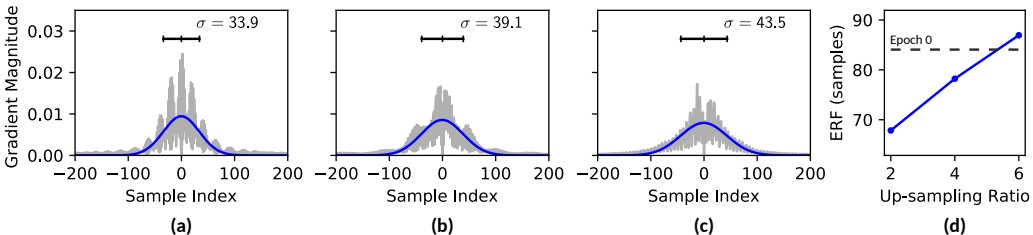

Figure 6: Measured gradient magnitude at the model input and associated Gaussian fit with *MU-GAN4* for **(a-c)** $R = 2,\ 4,\ 6$, respectively. **(d)** $2\sigma$ ERF trend, compared to ERF at initialization.

## 5 CONCLUSION

In this work we develop methods to enable the application of GANs to audio processing, in particular with classifier-free feature losses. In addition to several new model building blocks, we show that a convolutional autoencoder can be used to implement a high-performance feature loss in the context of audio super-resolution. Demonstrated on several speech and music super-resolution tasks, we show that our architecture achieves state-of-the-art performance in both objective and subjective metrics. We perform a detailed analysis of our model, and also show that effective receptive field size may be an important property that is not well-explored. Finally, our work raises new possibilities for the design and analysis of neural network-based synthesis methods in important problem domains beyond audio processing.

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

# A  APPENDIX

## A.1  MODEL PARAMETERS

## A.2  EVALUATION OF SUPERPIXEL LAYERS

We evaluate the impact of the superpixel layer proposed in Section 3 by comparing a baseline multiscale U-net with strided convolutions (*Strided*) against a multiscale U-net with superpixel layers (*Super*). We halve the number of convolutional kernels in each downsampling layer for *Super* such that the output feature map dimensions at each downsampling and upsampling layer are identical to those in *Strided*. Both model types are trained with the baseline $L2$ loss for 400 epochs on the *Speaker1* task.

Table 5: Comparison of superpixel and strided convolutional layers.

(a) Training time per minibatch

| | Depth Parameter | |
| --- | --- | --- |
| | $L = 4$ | $L = 8$ |
| *Strided* | 149.8 s | 195.1 s |
| *Super* | 128.1 s | 168.0 s |
| Speedup | 14.5% | 13.8% |

(b) Quality metrics

| | | Upsampling Ratio | | |
| --- | --- | --- | --- | --- |
| | | $R = 2$ | $R = 4$ | $R = 6$ |
| SNR | *Strided* | 21.67 | 18.45 | 14.91 |
| | *Super* | 21.75 | 18.41 | 14.89 |
| LSD | *Strided* | 1.67 | 2.20 | 2.73 |
| | *Super* | 1.70 | 2.08 | 2.39 |
| MOS-LQO | *Strided* | 3.57 | 3.13 | 2.71 |
| | *Super* | 3.53 | 3.15 | 2.78 |

## A.3  NOTE ON INTERMITTENT SPURIOUS NOISE

As described in Section 4, some audio clips super-resolved at $R = 4, 6$ exhibit intermittent, audible noise that is not present in the baseline reconstructions. We find that this noise is not inherent to our specific feature loss, and is present if we replace the unsupervised feature loss with a classifier-based feature loss. The spurious noise is also unrelated to the use of $\mathcal{L}_{adv}$, as it is present in audio generated by models trained only with $\mathcal{L}_f$. While a deeper investigation is planned for future work, we hypothesize that such noise stems from phase ambiguity in missing high-frequency content.

