# OpenReview forum: "Adversarial Audio Super-Resolution with Unsupervised Feature Losses"
_ICLR.cc/2019/Conference_

### Official Review · AnonReviewer2 · 2018-11-01
**nice work, confused about evaluation-related aspects**

**Rating:** 6
**Confidence:** 4

**Review:**

PRO’s:
+well-written
+nice overall system: GAN framework for super-sampling audio incorporating features from an autoencoder
+some good-sounding examples

CON’s:
-some confusing/weakly-presented parts (admittedly covering lots of material in short space)
-I am confused about the evaluation; would like additional qualitative/observational understanding of what works, including more on how the results differ from baseline

SUMMARY: The task addressed in this work is: given a low-resolution audio signal, generate corresponding high-quality audio. The approach is a generative neural network that operates on raw audio and train within a GAN framework.
Working in raw sample-space (e.g. pixels) is known to be challenging, so a stabilizing solution is to incorporate a feature loss. Feature loss, however, usually requires a network trained on a related task, and if such a net one does not already exist, then building one can have its own (possibly significant) challenges. In this work, the authors avoid this auxiliary challenge by using unsupervised feature losses, taking advantage of the fact that any audio signal can be downsampled, and therefore one has the corresponding upsampled signal as well.

The training framework is basically that of a GAN, but where, rather than providing the generator with a low-dimensional noise signal input, they provide the generator with the subsampled audio signal. The architecture includes a generator ( G(lo-fidelity)=high-fidelity ), a discriminator ( D(high-fidelity) = real or by super-sampled ? ), and an autoencoder ( \phi( signal x) = features of signal x at AE’s bottleneck).

COMMENTS:

The generator network appears to be nearly identical to that of Kuleshov et al (2017)-- which becomes the baseline-- and so the primary contribution differentiating this work is the insertion of that network into a GAN framework along with the additional feature-based loss term. This is overall a nice problem and a nice approach! In that light, I believe that there is a new focus in this work on the perceptual quality of the outputs, as compared to (Kuleshov et al 2017). I would therefore ideally like to see (a) some attempts at perceptually evaluating the resulting output (beyond PESQ, e.g. with human subjects and with the understanding that, e.g. not all AMT workers have the same aural discriminative abilities themselves), and/or (b) more detailed associated qualitative descriptions/visualization of the super-sampled signal, perhaps with a few more samples if that would help. That said, I understand that there are page/space limitations. (more on this next)

Given the similarity of the U-net architectures to (Kuleshov et al 2017), why not move some of those descriptions to the appendix?

For example, I found the description and figure illustrating the “superpixel layers” to be fairly uninformative: I see that the figure shows interleaving and de-interleaving, resulting in trading-off dimensionalities/ranks/etc, and we are told that this helps with well-known checkerboard artifacts, but I was confused about what the white elements represent, and the caption just reiterated that resolution was being increased and decreased. Overall, I didn’t really understand exactly the role that this plays in the system; I wondered if it either needed a lot more clarification (in an appendix?), or just less space spent on it, but keeping the pointers to the relevant references.  It seems that the subpixel layer was already implemented in Kuleshov 2017, with some explanation, yet in the present work a large table (Table 1(b)) is presented showing that there is no difference in quality metrics, and the text also mentions that there is no significant perceptual difference in audio. If the subpixel layer were explained in detail, and with justification, then I would potentially be OK with the negative results, but in this case it’s not clear why spend this time on it here. It’s possible that there is something simple about it that I am not understanding. I’m open to being convinced. Otherwise, why not just write: “Following (Kuleshov et al 2017), we use subpixel layers (Shi et al) [instead of ...] to speed up training, although we found that they make no significant perceptual effects.” or something along those lines, and leave it at that?

I did appreciate the descriptions of models’ sensitivity to size/structure of the conv filters, importance of the res connections, etc.

My biggest confusion was with the evaluation & results:

Since the most directly related work was (Kuleshov 2017), I compared the super resolution (U-net) samples on that website (https://kuleshov.github.io/audio-super-res/ ) to the samples provided for the present work ( https://sites.google.com/view/unsupervised-audiosr/home ) and I was a bit confused, because the quality of the U-net samples in (Kuleshov 2017) seemed to be perceptually significantly better than the quality of the Deep CNN (U-net) baseline in the present work. Perhaps I am in error about this, but as far as I can tell, the superresolution in (Kuleshov et al 2017) is significantly better than the Deep CNN examples here. Is this a result of careful selection of examples? I do believe what I hear, e.g. that the MU-GAN8 is clearly better on some examples than the U-net8. But then for non-identical samples, how come U-net4 actually generally sounds better than U-net8? That doesn’t make immediate sense either (assuming no overfitting etc). Is the benefit in moving from U-net4 to U-net8 within a GAN context but then stabilizing  it with the feature-based loss? If so, then how does MU-GAN8 compare to U-net4? Would there be any info for the reader by doing an ablation removing the feature loss from the GAN framework? etc. I guess I would like to get a better understanding of what is actually going on, even if qualitative. Is there any qualitative or anecdotal observation about which “types” of samples one system works better on than another? For example, in the provided examples for the present paper, it seemed to be the case that perhaps the MU-GAN8 was more helpful for supersampling female voices, which might have more high-frequency components that seem to get lost when downsampling, but maybe I’m overgeneralizing from the few examples I heard.

Some spectrograms might be helpful, since they do after all convey some useful information despite not telling much of the perceptual story. For example, are there visible but inaudible artifacts? Are such artifacts systematic?

Were individual audio samples represented as a one-hot encoding, or as floats? (I assume floats since there was no mention of sampling from a distribution to select the value).

A couple of typos:

descriminator → discriminator

pg 6 “Impact of superpixel layers” -- last sentence of 2nd par is actually not a sentence. “the reduction in convolutional kernels prior to the superpixel operation.”

Overall, interesting work, and I enjoyed reading it. If some of my questions around evaluation could be addressed-- either in a revision, or in a rebuttal (e.g. if I completely misunderstood something)-- I would gladly consider revising my rating (which is currently somewhere between 6 and 7).

---

> ### Author Response · Authors · 2018-11-26
> **Response to Reviewer #2**
>
>
> Thank you for the thoughtful and detailed review. Please see our enumerated responses below.
> We have also posted a comment above that summarized the changes in the latest revisio
>
> *Q1*: The generator network appears to be nearly identical to that of Kuleshov et al (2017)-- which becomes the baseline-- and so the primary contribution differentiating this work is the insertion of that network into a GAN framework along with the additional feature-based loss term. This is overall a nice problem and a nice approach! In that light, I believe that there is a new focus in this work on the perceptual quality of the outputs, as compared to (Kuleshov et al 2017). I would therefore ideally like to see (a) some attempts at perceptually evaluating the resulting output (beyond PESQ, e.g. with human subjects and with the understanding that, e.g. not all AMT workers have the same aural discriminative abilities themselves), and/or (b) more detailed associated qualitative descriptions/visualization of the super-sampled signal, perhaps with a few more samples if that would help. That said, I understand that there are page/space limitations. (more on this next)
>
> *A1*: Thank you for the detailed feedback. While time was short, we were able to add a subjective user study to the paper - see the updated Experiments section. We also added more qualitative discussion of super-sampled audio, with associated spectrograms. We are currently updating the web page with more audio samples as well and will update it shortly.
>
> *Q2*: Given the similarity of the U-net architectures to (Kuleshov et al 2017), why not move some of those descriptions to the appendix?
>
> *A2*: Thanks for this feedback - indeed, we found that our original architectural descriptions were overly detailed. We have removed much of this unnecessary detail, and plan to make a minor revision later today with additional architectural parameters in the appendix.
>
> *Q3*: Overall, I didn’t really understand exactly the role that [superpixel] plays in the system; I wondered if it either needed a lot more clarification (in an appendix?), or just less space spent on it, but keeping the pointers to the relevant references.  It seems that the subpixel layer was already implemented in Kuleshov 2017, with some explanation, yet in the present work a large table (Table 1(b)) is presented showing that there is no difference in quality metrics, and the text also mentions that there is no significant perceptual difference in audio. If the subpixel layer were explained in detail, and with justification, then I would potentially be OK with the negative results, but in this case it’s not clear why spend this time on it here. It’s possible that there is something simple about it that I am not understanding. I’m open to being convinced. Otherwise, why not just write: “Following (Kuleshov et al 2017), we use subpixel layers (Shi et al) [instead of ...] to speed up training, although we found that they make no significant perceptual effects.” or something along those lines, and leave it at that?
>
> *A3*: We concede that the presentation of superpixel and subpixel layers was somewhat confusing; we have revised the description in the Methods and Experiments sections to hopefully clarify this. One important detail we want to clarify is that while subpixel layers have been previously evaluated, no previous works have attempted to use its simple inverse to *decrease* spatial resolution (previous works use strided convolution or pooling). Our paper evaluates this somewhat obvious inverse operator, referred to as a superpixel layer, and finds that it actually reduces training time without loss of performance.
>
> *Q4*: Some spectrograms might be helpful, since they do after all convey some useful information despite not telling much of the perceptual story. For example, are there visible but inaudible artifacts? Are such artifacts systematic?
>
> *A4*: Indeed, while we were not able to include spectrograms in the initial draft, the revision includes spectrograms of super-resolved audio, as well as an example produced by a GAN that exhibits the artifacts we mentioned in the paper. To explicitly answer your question, we find that artifacts introduced by GANs are generally systematic (e.g., high-frequency whines), and both visible and audible.
>
> *Q5*: Were individual audio samples represented as a one-hot encoding, or as floats? (I assume floats since there was no mention of sampling from a distribution to select the value).
>
> *A5*: You are correct - the individual audio samples are 32-bit (single-precision) floats. We have added a short footnote in the Method section to clarify this. Thanks!
>
> *Q6*: A couple of typos: …
>
> *A6*: Thank you, these have all been corrected.

---

> > ### Author Response · Authors · 2018-11-27
> > **Response to Reviewer #2 (cont.)**
> >
> >
> > Q7: Since the most directly related work was (Kuleshov 2017), I compared the super resolution (U-net) samples on that website (https://kuleshov.github.io/audio-super-res/ ) to the samples provided for the present work ( https://sites.google.com/view/unsupervised-audiosr/home ) and I was a bit confused, because the quality of the U-net samples in (Kuleshov 2017) seemed to be perceptually significantly better than the quality of the Deep CNN (U-net) baseline in the present work. Perhaps I am in error about this, but as far as I can tell, the superresolution in (Kuleshov et al 2017) is significantly better than the Deep CNN examples here. Is this a result of careful selection of examples? I do believe what I hear, e.g. that the MU-GAN8 is clearly better on some examples than the U-net8. But then for non-identical samples, how come U-net4 actually generally sounds better than U-net8? That doesn’t make immediate sense either (assuming no overfitting etc). Is the benefit in moving from U-net4 to U-net8 within a GAN context but then stabilizing  it with the feature-based loss? If so, then how does MU-GAN8 compare to U-net4? Would there be any info for the reader by doing an ablation removing the feature loss from the GAN framework? etc. I guess I would like to get a better understanding of what is actually going on, even if qualitative. Is there any qualitative or anecdotal observation about which “types” of samples one system works better on than another? For example, in the provided examples for the present paper, it seemed to be the case that perhaps the MU-GAN8 was more helpful for supersampling female voices, which might have more high-frequency components that seem to get lost when downsampling, but maybe I’m overgeneralizing from the few examples I heard.
> >
> > A7: We suspect that the differences you mention may stem from discrepancies in training time. Specifically, Kuleshov et al., 2017 appear to train models for 400 epochs, while our models were trained for 150 epochs due to hardware and time constraints; it is certainly possible that our models did not reach full convergence, although we observed a practical plateau around 150 epochs. Thus, comparisons of model performance at our training levels should still be valid.
> >
> > We do have a comparison against U-net4 using the numbers published in Kuleshov et al., 2017 (see Table 1), however we didn’t find it was useful to do in-depth evaluation of this shallow version since early experiments showed that U-net8 was (unsurprisingly) better.
> >
> > Loss and model ablations have been added to the Experiments section, and indeed show that both depth and the proposed feature loss have significant impact on resulting performance.
> >
> > As you have suspected, we find that our model performs well on sounds that have more high-frequency content. While we aren’t entirely certain that our model always performs noticeably better with female voices, we do consistently find that consonant sounds are improved with our method. We have added a short discussion of this in the subjective quality analysis portion of the Experiments section.
> >
> > Thank you again for the thoughtful comments!

---

### Official Review · AnonReviewer3 · 2018-11-03
**Fascinating problem & fair results**

**Rating:** 5
**Confidence:** 4

**Review:**

This paper presents a GAN-based method to perform audio super-resolution. In contrast to previous work, this work uses auto-encoder to obtain feature losses derived from unlabeled data.

Comments:
(1) Redundant comma: “filters with very large receptive fields are required to create high quality, raw audio”.

(2) There are some state-of-the-art non-autoregressive generative models for audio waveform e.g., parallel wavenet, clarinet. One may properly discuss them in related work section. Although GAN performs very well for images, it hasn't obtained any compelling results for raw audios. Still, it’s very interesting to explore that. Any nontrivial insight would be highly appreciated.

(3) In multiscale convolutional layers, it seems only larger filter plays a significant role. What if we omit small filter, e.g., 3X1?

(4) It seems the proposed MU-GAN introduces noticeable noise in the upsampled audios.

Pros:
- Interesting idea and fascinating problem.
Cons:
- The results are fair. I didn’t see big improvement over previous work (Kuleshov et al., 2017).

I'd like to reconsider my rating after the rebuttal.

---

> ### Author Response · Authors · 2018-11-26
> **Response to Reviewer #3**
>
>
> We thank the reviewer for the feedback. Please see our enumerated responses below.
> We have also posted a comment above that summarized the changes in the latest revision.
>
> *Q1*: Redundant comma: “filters with very large receptive fields are required to create high quality, raw audio”.
>
> *A1*: We have corrected this, thank you.
>
> *Q2*: There are some state-of-the-art non-autoregressive generative models for audio waveform e.g., parallel wavenet, clarinet. One may properly discuss them in related work section. Although GAN performs very well for images, it hasn't obtained any compelling results for raw audios. Still, it’s very interesting to explore that. Any nontrivial insight would be highly appreciated.
>
> *A2*: Thanks for pointing this out. We have added a discussion of these works, and a comparison to other GAN-based methods in the Related Works section.
>
> *Q3*: In multiscale convolutional layers, it seems only larger filter plays a significant role. What if we omit small filter, e.g., 3X1?
>
> *A3*: This is a good point - we found that small filters do play a marginal but noticeable role in audio quality and speed of convergence. We suspect that while smaller filters are comparatively less powerful, they are easier to optimize and may have an important role in fitting many of the “easy” components in audio signals. Smaller filters also have lower overhead in terms of computational requirements, making our networks (which have hundreds of feature maps in some layers) feasible to train in a few days or less with a single GPU.
>
> *Q4*:  It seems the proposed MU-GAN introduces noticeable noise in the upsampled audios.
>
> *A4*: Thanks for this comment - we noticed this as well. We have copied our answer A4 to reviewer #1 below:
> We did notice that in some samples, especially at higher upsampling rates, there are instances of noise on utterances with significant high-frequency content (e.g., fricatives and aspiration). We are not entirely certain on the cause of this noise, but we suspect that it is related to inherent ambiguity in the phase and magnitude of high frequency signals. Furthermore, we found that this noise is present even if we replace the unsupervised feature loss with other conventional feature losses. Nevertheless, we made sure to include samples at high up-sampling ratios that included this noise in the user study, which indicated that users preferred audio produced by our method in spite of spurious noise. We have added a note in the paper regarding this problem.
>
> *Q5*: The results are fair. I didn’t see big improvement over previous work (Kuleshov et al., 2017).
>
> *A5*: We appreciate the criticism. We wanted to highlight that while our baselines are based on the work from Kuleshov et al., 2017, our primary baseline for evaluation (U-net8) is a much deeper and more powerful model compared to the model evaluated in Kuleshov et al., 2017. For a more direct comparison with Kuleshov et al., 2017, see the results for U-net4 in the Experiments section, which are taken from the authors’ paper.

---

### Official Review · AnonReviewer1 · 2018-11-04

**Rating:** 4
**Confidence:** 4

**Review:**


The paper presents a model to perform audio super resolution. The proposed model trains a neural network to produce a high-resolution audio sample given a low resolution input. It uses three losses: sample reconstructon, adversarialy loss and feature matching on a representation learned on an unsupervised way.

From a technical perspective, I do not find the proposed approach very novel. It uses architectures following closely what has been done for Image supre-resolution. I am not aware of an effective use of GANs in the audio processing domain. This would be a good point for the paper. However, the evidence presented does not seem very convincing in my view. While this is an audio processing paper, it lacks domain insights (even the terminology feels borrowed from the image domain). Again, most of the modeling decisions seem to follow what has been done for images. The empirical results seem good, but the generated audio does not match the quality of the state-of-the-art.

The presentation of the paper is correct. It would be good to list or summarize the contributions of this work.

Recent works have shown the amazing power of auto-regressive generative models (WaveNet)  in producing audio signals. This is, as far as I know, the state-of-the-art in audio generation. The authors should motivate why the proposed model is better or worth studying in light of those approaches. In particular, a recent work [A] has shown very high quality results in the problem of speech conversion (which seems harder than bandwidth extension). It would seem to me that applying such models to the bandwith extension task should also lead to very high quality results as well. What is the advantage of the proposed approach? Would a WaveNet decoder also be improved by including these auxiliary losses?

While the audio samples seem to be good, they are also a bit noisy even compared with the baseline. This is not the case in the samples generated by [A] (which is of course a different problem).

The qualitative results are evaluated using PESQ. While this is a good proxy it is much better to perform blind tests with listeners. That would certainly improve the paper.

Feature spaces are used in super resolution to provide a space in which the an L2 loss is perceptually more relevant. There are many such representations for audio signals. Specifically the magnitude of time-frequency representations (like spectrograms) or more sophisticated features such as scattering coefficients. In my view, the paper would be much stronger if these features would be evaluated as alternative to the features provided by the proposed autoencoder.

One of the motivations for defining the loss in the feature space is the lack (or difficulty to train) auxiliary classifiers on large amounts of data.  However, speech recognition models using neural networks are quite common. It would be good to also test features obtained from an off-the-shelf speech recognition system. How would this compare to the proposed model?

The L2 "pixel" loss seems a bit strange in my view. Particularly in audio processing, the recovered high frequency components can be synthesized with an arbitrary phase. This means that imposing an exact match seems like a constraint as the phase cannot be predicted from the low resolution signal (which is what a GAN loss could achieve).

The paper should present ablations on the use of the different losses. In particular, one of the main contributions is the inclusion of the loss measured in the learned feature space. The authors mention that not including it leads to audible artifacts. I think that more studies should be presented (including quantitative evaluations and audio samples).

How where the hyper parameters chosen? is there a lot of sensitivity to their values?


[A] van den Oord, Aaron, and Oriol Vinyals. "Neural discrete representation learning." Advances in Neural Information Processing Systems. 2017.

---

> ### Author Response · Authors · 2018-11-26
> **Response to Reviewer #1 (1/2)**
>
>
> We thank the reviewer for the thoughtful and detailed response. Please see our enumerated responses below.
> We have also posted a comment above that summarized the changes in the latest revision.
>
> *Q1*: From a technical perspective, I do not find the proposed approach very novel. It uses architectures following closely what has been done for Image super-resolution. I am not aware of an effective use of GANs in the audio processing domain. This would be a good point for the paper. However, the evidence presented does not seem very convincing in my view. While this is an audio processing paper, it lacks domain insights (even the terminology feels borrowed from the image domain). Again, most of the modeling decisions seem to follow what has been done for images. The empirical results seem good, but the generated audio does not match the quality of the state-of-the-art.
>
> *A1*: Thank you for this feedback. We agree that some of our high-level design choices are inspired by architectures from image processing literature. However, the main focus of our work is the exploration of such techniques and their adaptations to audio processing, which has been little-explored previously. Most importantly, we develop several new techniques and present analysis that is found in neither image nor audio processing literature. Indeed, autoregressive methods produce audio of excellent quality; we have added a discussion of this to the paper, and also elaborate more in this topic in answer A3 below.
>
> *Q2*: The presentation of the paper is correct. It would be good to list or summarize the contributions of this work.
>
> *A2*: We agree - the introduction has been revised and now includes an explicit list of contributions.
>
> *Q3*: Recent works have shown the amazing power of auto-regressive generative models (WaveNet)  in producing audio signals. This is, as far as I know, the state-of-the-art in audio generation. The authors should motivate why the proposed model is better or worth studying in light of those approaches. In particular, a recent work [A] has shown very high quality results in the problem of speech conversion (which seems harder than bandwidth extension). It would seem to me that applying such models to the bandwith extension task should also lead to very high quality results as well. What is the advantage of the proposed approach? Would a WaveNet decoder also be improved by including these auxiliary losses?
>
> *A3*: We agree and note in the paper that auto-regressive models are indeed a promising avenue for audio generation. The primary difference from our work is that auto-regressive methods require an inference pass to generate a single output sample, with an input sequence that grows with each inference inference pass. This process is computationally intensive - for instance, at 16 KHz, an optimized Wavenet requires ~1.5 minutes to generate one second of audio [C]. We were recently made aware of a Wavenet variant that alleviates the issue of slow sample generation with a model distillation/student-teacher method [B]. As you correctly point out, Wavenet models can be improved with these auxiliary losses, and the Wavenet variant in [B] actually integrates a feature loss based on a speech phoneme-classifier network. This supports our view that our work is not in conflict with Wavenet and other auto-regressive methods, but rather augments them and can be used successfully in conjunction.
>
> *Q4*: While the audio samples seem to be good, they are also a bit noisy even compared with the baseline. This is not the case in the samples generated by [A] (which is of course a different problem).
>
> *A4*: We did notice that in some samples, especially at higher upsampling rates, there are instances of noise on utterances with significant high-frequency content (e.g., fricatives and aspiration). We are not entirely certain on the cause of this noise, but we suspect that it is related to inherent ambiguity in the phase and magnitude of high frequency signals. Furthermore, we found that this noise is present even if we replace the unsupervised feature loss with other conventional feature losses. We made sure to include samples at high up-sampling ratios that included this noise in the user study, and the study indicated that users preferred audio produced by our method in spite of spurious noise. Nevertheless we have added a note in the paper regarding this problem.
>
> *Q5*: The qualitative results are evaluated using PESQ. While this is a good proxy it is much better to perform blind tests with listeners. That would certainly improve the paper.
>
> *A5*: Thanks for this point. We have acted on this recommendation and now have results from a user study in the Experiments section.
>
> [B] van Den Oord et al. “Parallel WaveNet: Fast High-Fidelity Speech Synthesis.” ICML 2018.

---

> > ### Author Response · Authors · 2018-11-26
> > **Response to Reviewer #1 (2/2)**
> >
> >
> > *Q6*: Feature spaces are used in super resolution to provide a space in which the an L2 loss is perceptually more relevant. There are many such representations for audio signals. Specifically the magnitude of time-frequency representations (like spectrograms) or more sophisticated features such as scattering coefficients. In my view, the paper would be much stronger if these features would be evaluated as alternative to the features provided by the proposed autoencoder.
> >
> > *A6*: Thank you for this point. We actually have run experiments similar to the ones the reviewer mentions, but found the results to be either worse or no different to a standalone L2 loss and abandoned subsequent efforts. Specifically, we experimented with L2 losses in the Fourier transform space, and losses across the coefficients of various wavelet transforms. In general, we don’t expect linear transforms to perform better than or on par compared to non-linear transforms similar to those presented in this paper. A classic but effective example is the comparison of linear PCA to the non-linear PCA provided by a single-layer autoencoder.
> >
> > *Q7*: One of the motivations for defining the loss in the feature space is the lack (or difficulty to train) auxiliary classifiers on large amounts of data.  However, speech recognition models using neural networks are quite common. It would be good to also test features obtained from an off-the-shelf speech recognition system. How would this compare to the proposed model?
> >
> > *A7*: We agree that off-the-shelf speech recognitions systems are an important comparison point. We were able to find a speech classifier-based feature loss used for speech denoising, and added the analysis to paper. We found that our method performed on-par (and better in some cases) compared to the classifier-based loss. Besides the competitive quantitative performance of our work, our methods generalize to different types of audio beyond speech. Finding classifier models for every type of audio is fraught with practical issues and may not be generally feasible. E.g., for music, a variety of different classification granularities and types exist, and it’s not clear what kind of classifier should be selected. Our method provides users with a way to train models with a feature loss that matches the characteristics of their own dataset, regardless of the audio type and how (or if) their dataset may be labeled.
> >
> > *Q8*: The L2 "pixel" loss seems a bit strange in my view. Particularly in audio processing, the recovered high frequency components can be synthesized with an arbitrary phase. This means that imposing an exact match seems like a constraint as the phase cannot be predicted from the low resolution signal (which is what a GAN loss could achieve).
> >
> > *A8*: We absolutely agree. Indeed, we tried other methods for a baseline loss such as 1-dimensional analogues of “texture losses” that use correlations instead of exact match metrics, but couldn’t achieve good results. Note that your observation is exactly what our results indicate - that deviating from an exact match (as shown by lower SNR) can yield better results in terms of perceptual quality. We do see potential in methods that relax phase constraints further, for instance the phase shuffle operation from [C], but considered it an orthogonal technique that was out of scope for this paper.
> >
> > *Q9*: The paper should present ablations on the use of the different losses. In particular, one of the main contributions is the inclusion of the loss measured in the learned feature space. The authors mention that not including it leads to audible artifacts. I think that more studies should be presented (including quantitative evaluations and audio samples).
> >
> > *A9*: Thanks for this comment - we have added these details to the paper (e.g., see the model ablation study in the Experiments section). We are currently adding more samples to the webpage and will update it shortly.
> >
> > *Q10*: How where the hyper parameters chosen? is there a lot of sensitivity to their values?
> >
> > *A10*: Hyperparameters were typically determined with small parameter sweeps over generally accepted values (e.g., batch size of 32-64). However, some model parameters (such as network depth) were somewhat constrained by training time. Except for depth, we didn’t find significant differences from our parameter sweeps, including those used for the optimizer. We will make sure to include this information and more to the Appendix in a later revision.
> >
> > [C] Chris Donahue, Julian McAuley, and Miller Puckette. “Synthesizing audio with generative adversarial networks.” ICLR Workshops. 2018.

---

### Author Response · Authors · 2018-11-26
**Summary of revision**

We appreciate the reviewers’ detailed feedback and thoughtful questions. We have responded to each review independently, and summarize changes to the manuscript here.

Major changes include the addition of a qualitative user study, a model ablation analysis, comparisons against an off-the-shelf speech classifier-based feature loss, and an effective receptive field analysis. We have also added more spectrograms of super-resolved audio signals, and have added many new samples to the web page. Note that the webpage has moved to https://mugandemo.github.io/mugandemo .

Other changes related to writing and clarity include the addition of a list of contributions, general writing improvements and typo fixes, and some additional discussion of autoregressive methods in the Related Works section.

We also note that due to a bug in our python audio processing scripts, MOS-LQO scores from the old draft were truncated. We have updated all relevant results tables with the correct, full-range MOS-LQO metrics. Note that while the absolute numbers have changed somewhat, the trends and conclusions drawn from the old results are still valid.

Again, thank you for the helpful feedback, it is greatly appreciated!

---

### Public Comment · ~Praveen_Narayanan1 · 2019-05-30
**Code for paper**

Do the authors have a usable implementation we could play with?

---

### Meta-Review · Area_Chair1 · 2018-12-14
**interesting approach, but results are not compelling enough**

**Confidence:** 5
**Recommendation:** Reject

**Metareview:**

The paper presents an algorithm for audio super-resolution using adversarial models along with additional losses, e.g. using auto-encoders and reconstruction losses, to improve the generation process.

Strengths
- Proposes audio super resolution based on GANs, extending some of the techniques proposed for vision / image to audio.
- The authors improved the paper during the review process by including results from a user study and ablation analysis.

Weaknesses
- Although the paper presents an interesting application of GANs for the audio task, overall novelty is limited since the setup closely follows what has been done for vision and related tasks, and the baseline system. This is also not the first application of GANs for audio tasks.
- Performance improvement over previously proposed (U-Net) models is small. It would have been useful to also include UNet4 in user-study, as one of the reviewers’ pointed out, since it sounds better in a few cases.
- It is not entirely clear if the method would be an improvement of state-of-the-art audio generative models like Wavenet.

Reviewers agree that the general direction of this work is interesting, but the results are not compelling enough at the moment for the paper to be accepted to ICLR. Given these review comments, the recommendation is to reject the paper.